# Advances in Protein Kinase Regulation of Stress Responses in Fruits and Vegetables

**DOI:** 10.3390/ijms26020768

**Published:** 2025-01-17

**Authors:** Yanan Song, Fujun Li, Maratab Ali, Xiaoan Li, Xinhua Zhang, Zienab F. R. Ahmed

**Affiliations:** 1College of Agricultural Engineering and Food Science, Shandong University of Technology, Zibo 255000, China; synnet163@163.com (Y.S.); lifujun@sdut.edu.cn (F.L.); maratab_ali@sdut.edu.cn (M.A.); lixa@sdut.edu.cn (X.L.); 2School of Food and Agricultural Sciences, University of Management and Technology, Lahore 54000, Pakistan; 3Integrative Agriculture Department, College of Agriculture and Veterinary Medicine, United Arab Emirates University, Al Ain 15551, United Arab Emirates

**Keywords:** protein kinase, F&Vs, abiotic stress, disease resistance, regulation mechanism

## Abstract

Fruits and vegetables (F&Vs) are essential in daily life and industrial production. These perishable produces are vulnerable to various biotic and abiotic stresses during their growth, postharvest storage, and handling. As the fruit detaches from the plant, these stresses become more intense. This unique biological process involves substantial changes in a variety of cellular metabolisms. To counter these stresses, plants have evolved complex physiological defense mechanisms, including regulating cellular activities through reversible phosphorylation of proteins. Protein kinases, key components of reversible protein phosphorylation, facilitate the transfer of the γ-phosphate group from adenosine triphosphate (ATP) to specific amino acid residues on substrates. This phosphorylation alters proteins’ structure, function, and interactions, thereby playing a crucial role in regulating cellular activity. Recent studies have identified various protein kinases in F&Vs, underscoring their significant roles in plant growth, development, and stress responses. This article reviews the various types of protein kinases found in F&Vs, emphasizing their roles and regulatory mechanisms in managing stress responses. This research sheds light on the involvement of protein kinases in metabolic regulation, offering key insights to advance the quality characteristics of F&Vs.

## 1. Introduction

Fruits and vegetables (F&Vs) are essential components of our diet, playing crucial roles in daily consumption and industrial production. However, recent industrialization and climate change have increasingly exposed F&Vs to various abiotic stresses, such as salt stress, drought, high temperatures, low temperatures, and biotic stresses, including viruses, microorganisms, and pests [1]. These stresses significantly impact F&Vs’ physiology, impeding their growth and development and ultimately reducing productivity [2]. For example, excessive salt in the soil can lead to ion toxicity and suppress photosynthesis in F&Vs, thereby limiting their growth and development. Approximately 7% of global soils are impacted by salinization [3]. Drought negatively affects nutrient absorption and yield, with global losses estimated at USD 166 billion over the past 27 years [4]. Climate change has also significantly influenced the growth and development of F&Vs. The continuous rise in global temperatures, with an estimated 0.3 °C rise in surface temperatures over the next decade, poses challenges such as delayed seed germination, disrupted fruit set, and various induced morphological and physiological changes, ultimately resulting in decreased productivity [5,6]. Under low-temperature conditions, cold-sensitive F&Vs are prone to damage such as skin indentation, flesh browning, and delayed ripening [7]. Diseases, particularly pathogenic infections, significantly threaten the growth and quality of F&Vs, compromising not only the quality of the produce but also consumer health [8]. To withstand these stresses, F&Vs have developed various regulatory mechanisms over long-term evolution. Therefore, understanding stress tolerance mechanisms is crucial for ensuring optimal growth and yield of F&Vs. This knowledge will aid in the development of stress-tolerant varieties, thereby promoting the stability and sustainability of horticultural crops.

Reversible phosphorylation of proteins is a crucial post-translational modification process and a widely recognized regulatory mechanism involved in nearly all physiological and pathological processes [9]. Recent studies have highlighted the key regulatory roles of this process in the growth, development, and stress responses of F&Vs. Protein kinases, integral to reversible protein phosphorylation, catalyze the transfer of the γ-phosphate group from adenosine triphosphate (ATP) to the side chains of serine (Ser), threonine (Thr), and tyrosine (Tyr) on substrates, initiating protein phosphorylation reactions [10]. This process leads to structural changes in proteins that modulate their activity. Through mitogen-activated protein kinase (MAPK) cascades, protein phosphorylation can transmit and amplify external signals by modulating gene expression through the regulation of transcription factors, ion channels, and other pathways [11]. Many protein kinases have been identified in various F&Vs, demonstrating their involvement in the growth, development, and stress responses in plants through various mechanisms. However, a comprehensive overview of the types of protein kinases in the F&Vs and their regulatory roles in stress responses is still lacking. Thus, this article introduces the types of protein kinases in F&Vs, emphasizing their roles and regulatory mechanisms in stress responses and offering valuable insights on the metabolic regulation of F&Vs and the enhancement of their quality characteristics.

## 2. Classification of Protein Kinases

Protein kinases represent a vast family of plant proteins that play diverse roles and contribute to critical structural modifications. Consequently, classification of protein kinases aids in understanding their structures and functions. Initially, protein kinases were classified into Ser/Thr and Tyr kinases based on their substrate specificity for phosphorylating Ser, Thr, and Tyr residues, respectively. This classification was further refined by Lehti-Shiu and Shiu [12], who identified protein kinases from 25 plant genomes and grouped them into the following categories: AGC (which includes cAMP-dependent protein kinase A (PKA), cGMP-dependent protein kinase G (PKG), and phospholipid-dependent protein kinase C (PKC)), calcium/calmodulin-regulated kinases (CaMK), CMGC (which includes cyclin-dependent kinases (CDKs), mitogen-activated protein kinases (MAPKs), glycogen synthase kinases (GSKs), and CDK-like kinases (CLKs)), casein kinase 1 (CK1), sterility (STE), tyrosine kinase (TK), tyrosine kinase-like kinases (TKL), and others.

### 2.1. AGC

AGC kinases are characterized by their Ser/Thr protein kinase nature and sequence similarity to the catalytic kinase domains of PKA, PKG, and PKC [13]. PKA functions as a tetrameric enzyme composed of two regulatory subunits and two catalytic subunits, and is further classified into PKA type I and type II based on distinct regulatory subunits. The PKA regulatory subunits form dimers through interactions at their N-terminus dimerization/docking domains [14]. Among the three types of PKA catalytic subunits (Cα, Cβ, and Cγ), Cα and Cβ are the dominant forms with multiple splice variants [15]. Plant PKGs exhibit a unique structure, encompassing both protein kinase and phosphatase domains, with a type 2C protein phosphatase domain that distinguishes them from those found in animals [16]. PKC is categorized into three groups based on activator sensitivity: conventional PKC (including PKCα, PKCβ, and PKCγ), novel PKC (including PKCδ, PKCε, PKCη, and PKCθ), and atypical PKC (including of PKCζ and PKCλ/ɩ) [17]. Structurally, the PKC polypeptide chain features four conserved domains (C1–C4). The N-terminus regulatory domain (comprising C1 and C2) regulates kinase activity and subcellular localization, whereas the C-terminus catalytic domain (comprising C3 and C4) binds to ATP and substrate proteins [18]. Recent studies have demonstrated the crucial role of AGC kinases in various F&Vs, including tomato (Solanum lycopersicum) [19] and bean (Phaseolus vulgaris) [20]. These kinases are particularly important for phosphorylating auxin efflux carriers, which affects plant growth and development by modulating auxin transport [21].

### 2.2. CaMK

CaMK includes calcium-dependent protein kinase (CDPK) and sucrose non-fermenting 1-related protein kinase (SnRK) families. CDPK is a monomeric protein with a molecular mass ranging from 40 to 90 kDa [22] and consists of four distinct domains: N-terminus and C-terminus variable domains, a Ser/Thr kinase domain, an autoinhibitory junction, and a regulatory calmodulin-like domain with Ca^2+^-binding EF-hands [23]. CDPK plays a crucial role as a receptor for Ca^2+^ signal transduction in plant cells due to its Ca^2+^-binding activity [24]. SnRKs are categorized into three subfamilies: SnRK1, SnRK2, and SnRK3. SnRK1 features a highly conserved N-terminus protein kinase domain, while SnRK2 and SnRK3, which are unique to plants, have a highly conserved N-terminus kinase domain and a C-terminus variable regulatory domain in SnRK2 [25]. SnRK3, also known as calcineurin B-like protein-interacting protein kinases (CIPKs), includes N-terminus protein kinase domains and a C-terminus regulatory domain (NAF domain) [26]. Numerous CaMKs have been identified in various F&Vs (Table 1).

### 2.3. CMGC

CMGC kinases derive their name from CDKs, MAPKs, GSKs, and CLKs. CDKs typically comprise 200–350 amino acid residues, with a molecular weight of approximately 30–45 kDa. Structurally, they are characterized by a β-folded amino-terminal region and an α-helical C-terminus domain [69]. CDKs can be further divided into CDKA–CDKG and CDKLIKE, with CDKAs and CDKBs primarily involved in cell cycle regulation [70]. Various CDKs are activated at specific times during the cell cycle, driving cells through the cell cycle by phosphorylating substrates. In addition to regulating the cell cycle, CDKs are involved in transcription and messenger RNA (mRNA) processing [71]. The MAPK cascades, including MAPK, MAPK kinase (MAPKK), and MAPKK kinase (MAPKKK), constitute highly conserved signal transduction pathways in all eukaryotes. Generally, the MAPK pathway involves a phosphorylation cascade, starting with MAPKKK acting on MAPKK, which then phosphorylates MAPK, leading to sequential signal transduction and specific physiological and biochemical responses [72]. Numerous MAPKs have been identified in F&Vs (Table 1). GSKs, a type of Ser/Thr protein kinase, are encoded by a multigene family in plants and are involved in various processes of plant growth and development [73]. CLKs mainly consist of four subtypes CLK1-4 and are known as “LAMMER” kinases due to a unique conserved amino acid sequence, “EHLAMMERILG”, found in their catalytic domain. The differences among them lie in the variations in the ATP-binding sites and their respective functions. CLKs phosphorylate Tyr and Ser/Thr residues, modulate the physiological properties of protein substrates, and play crucial roles in regulating pre-mRNA splicing and signal transduction processes [74].

### 2.4. CK1

The CK1 family is evolutionarily conserved across eukaryotes and is characterized by a highly conserved catalytic domain and a variable domain primarily located at the C-terminus [75]. CK1 plays pivotal roles in various biological processes, such as DNA damage response, cytokinesis, cell cycle regulation, apoptosis, immune responses, biotic stress responses, and flowering time regulation [76,77]. However, the specific biological functions of CK1 members in F&Vs remain largely unknown and most of the CK1 family members have yet to be identified.

### 2.5. STE

STE kinases include STE7, STE11, and STE20, which are upstream regulators of MAPKs [78]. In the signal transduction pathway, STE7, STE11, and STE20 function as MAPKK, MAPKKK, and MAPKKKK, respectively [79,80]. The STE7 family directly phosphorylates MAPK, the STE11 family phosphorylates STE7 kinases, and many STE20 members target STE11 kinases [81].

### 2.6. TK

TKs are a large family of kinases categorized as receptor tyrosine kinases and non-receptor tyrosine kinases, based on their localization in the cell membrane [82]. These kinases catalyze the transfer of the γ-phosphate from ATP to protein tyrosine residues. A defining characteristic is the typical TK domain at their carboxyl terminus, which possesses a core catalytic structure capable of self-phosphorylation, and substrate phosphorylation. However, each TK contains unique domains, motifs, and residues that confer distinct catalytic or regulatory properties [83].

### 2.7. TKL

TKLs are a diverse group of Ser/Thr protein kinases that resemble TKs in sequence but lack the distinct motifs found in tyrosine kinases. They represent the most abundant kinase group in plant, accounting for approximately 80% of the plant kinome [84].

## 3. Regulation of Protein Kinase in F&V Stress Responses

### 3.1. The Role of Protein Kinases in the Salt Stress Response of F&Vs

Salt stress is a prevalent abiotic stress factor that significantly impacts the growth and development of F&Vs, adversely affecting various aspects, and ultimately compromising growth, development, and yield [85]. Several studies using transcriptome analysis have demonstrated that protein kinases are the key components of signal transduction pathways and play crucial roles in the salt stress response of F&Vs. For instance, Wei et al. [86] identified approximately 42 differentially expressed genes enriched in the MAPK signaling pathway in tomato seedlings in response to salt stress. Inhibition of MAPK phosphorylation with SB203580 reduced the levels of hormones, such as jasmonic acid (JA) and abscisic acid (ABA), as well as inhibited the activities of antioxidant enzymes, indicating MAPK’s role in the salt stress response of tomato seedlings. Irrigating tomato seedlings with 100 mmol L^−1^ NaCl solution for 7 days demonstrated that the overexpression of *SlMAPK3* alleviated the inhibition of salt stress on the fresh weight, seedling height, seminal root length and seed germination of tomato plants. And the overexpression of *SlMAPK3* in tomatoes enhanced the clearance of reactive oxygen species (ROS), and upregulated the expression of genes associated with the ethylene signaling pathway, thereby improving tomato tolerance to salt stress [87]. After 10 days of cultivating grape callus on MS medium containing NaCl, it was found that the overexpressed *VvMAPK9* grape callus grew faster and exhibited lower electrical conductivity compared to the wild-type grape callus. Thus, overexpression of *VvMAPK9* in grapes increased callus tolerance to salt stress [32]. Additionally, the response of F&Vs to salt stress is significantly regulated by CDPKs. Under salt stress conditions, overexpression of *VaCPK21* in cultured grape cells and its heterologous expression in *Arabidopsis* seedlings led to a significant upregulation of resistance-related genes and enhanced salt stress resistance [88]. The salt overly sensitive (SOS) pathway is a well-known salt resistance signaling pathway in plants, in which calcineurin B-like proteins (CBLs) serve as Ca^2+^ receptors to specifically activate CIPKs. Studies have demonstrated that overexpression of *SlCBL4* or *SlCIPK24* [89], as well as heterologous expression of the CBL-CIPK gene *MdSOS2L1* in tomatoes, can enhance salt tolerance [90]. Furthermore, the heterologous expression of peach *PpSnRK1a* in tomatoes could enhance the ABA signal transduction system, ROS metabolism, and leaf lipid peroxidation, thereby increasing the plant’s ability to withstand salt stress [91]. These studies highlight the significant role of protein kinases in mediating the response of F&Vs to salt stress (Figure 1), providing valuable insights into the mechanisms underlying salt stress in F&Vs and strategies for enhancing their salt tolerance.

### 3.2. The Role of Protein Kinases in the Temperature Stress Response of F&Vs

Temperature plays a crucial role in the growth and development of F&Vs. High temperatures can impede normal growth, increase transpiration, weaken photosynthesis, and, in severe cases, cause dehydration symptoms that may lead to plant death [92]. Conversely, prolonged exposure to low temperatures can damage plant cells, causing chilling injuries or frost damage [93].

Protein kinases play a crucial role in the perception and transduction of temperature signals in F&Vs. For example, lettuce treated under high-temperature conditions (33 ± 2 °C daytime/25 ± 2 °C night) showed that silencing of *LsMAPK4* resulted in slower stem growth and significantly inhibited accelerated bolting, indicating that LsMPAK4 might be a potential regulator of bolting in lettuce and could promote bolting under high-temperature [94]. In tomatoes, studies have demonstrated that knocking out *SlMPK3* enhances tolerance to high temperatures [44], while overexpression of *SlMPK3* inhibits plant growth and reduces the activity of antioxidant enzymes, indicating that SlMPK3 negatively regulates tomato’s high-temperature resistance. Additionally, the knockout of the CDPK gene *CPK28* in tomatoes led to ROS accumulation, protein oxidation, and decreased antioxidant enzyme activity under high-temperature stress, suggesting a positive role for CPK28 in the tomato plant’s response to high temperatures [95]. Under low-temperature stress, overexpression of the necrotic dwarf gene (*NDW*), a receptor-like protein kinase gene, was found to reverse the semi-dwarf and necrotic phenotypes induced by low temperatures in tomato plants, highlighting the crucial role of *NDW* in tomato resistance to low temperatures [96]. Furthermore, after 8 days of low temperature culture, heterologous overexpression of *SpCPK33* in tomato plants showed stronger cold resistance and reduced leaf wilting by decreasing the malondialdehyde content and ROS levels compared with wild-type plants [97]. In bananas, overexpression of *MusaMPK5* promoted stem growth, increased fresh weight, and enhanced resistance to low temperatures [98]. In peaches, 17 *PpCDPKs* were identified in the genome, with significant changes in the expression of most *PpCDPKs* observed during cold storage. Transcriptome analysis revealed that *PpCDPK2*, *PpCDPK7*, *PpCDPK10*, and *PpCDPK13* are closely associated with postharvest low-temperature stress in peaches [37]. Overall, these findings underscore the important roles of protein kinases in the responses of various F&Vs to both high and low-temperature stresses, as illustrated in Figure 1.

### 3.3. The Role of Protein Kinases in the Drought Tolerance of F&Vs

Drought stress significantly impacts the growth, development, and yield of F&Vs. Zhu et al. [99] reported that using CRISPR-Cas9 to generate *CPK27*-knocked out tomato lines and suspending water supply for 10 days to simulate drought stress resulted in severe leaf atrophy in the knockout plants, indicating that CPK27 positively regulates plant drought tolerance in tomato plant. Furthermore, overexpression of *CaDIK1* [100], *CaCIPK3* [101], and *CaCIPK7* [102] could enhance the antioxidant capacity and drought tolerance of pepper. In tomatoes, overexpression of *SlMAPK3* improved the photosynthetic capacity of leaves and increases the content of chlorophyll, proline, and sugar, thereby improving drought tolerance [103]. Under drought stress, *SlMAPK3*-overexpressing tomato exhibited higher relative water content, SOD content, lower POD content, and stronger drought tolerance [104]. VaCIPK02 regulated drought stress in grapes by interacting with ABA receptor proteins and modulating ABA accumulation [105]. These findings highlight the crucial regulatory role of protein kinases in the drought resistance mechanisms of F&Vs (Figure 1).

### 3.4. The Role of Protein Kinases in Disease Resistance of F&Vs

F&Vs are vulnerable to various pests and pathogenic microorganisms throughout their growth, development, and postharvest stages [106]. In response to these challenges, F&Vs have developed diverse defense mechanisms. Notably, protein kinases have been a key focus of research aimed at understanding their role in the disease resistance mechanisms of F&Vs.

CDPK and CDPK-related kinases (CRK) play various roles in plant defense. For instance, SlCDPK18 and SlCDPK10 positively regulated tomato resistance to *Xanthomonas oryzae* pv. *oryzae* and *Pseudomonas oryzae syringae* pv. *tomato* (*Pst*) DC3000. Similarly, SlCRK6 enhanced tomato resistance to *Pst* DC3000 and *Sclerotinia sclerotiorum* [22]. Protein kinases have also been extensively investigated for their role in fungal resistance in F&Vs. *Botrytis cinerea*, the causative agent of gray mold, severely affects tomatoes and poses threats during their growth, development, and postharvest stages [107]. Research has identified tomato protein kinase 1b (TPK1b) as a crucial regulator in tomato defense against gray mold. TPK1b could be phosphorylated by the receptor-like kinase 1 (PORK1), which was an ortholog of tomato PEPR1/2. The inhibition of PORK1 resulted in decreased phosphorylation of TPK1b, thereby compromising tomato resistance to gray mold [108]. Additionally, *NDW* and SlMAPKKK43 played crucial positive regulatory roles in tomato resistance to gray mold. When isolated leaves and fruits of transgenic tomato lines were inoculated with *Botrytis cinerea*, it was found that the lesion area in *SlMAPKKK43* knocked out plants was significantly larger than that in wild-type plants. Phosphorylation validation revealed that SlMAPKKK43 positively regulated tomato resistance to gray mold by phosphorylating SlMKK2 and SlMKK4 [109]. In pears, 108 PbrMAPKKKs were identified, with the expression of *PbrMAPKKK12*/*13*/*53*/*60*/*65*/*82*/*83*/*96* positively correlating with resistance to pear black spot disease. Conversely, silencing *PbrMAPKKK6* resulted in increased resistance to black spot disease in pears [110]. In muskmelons, infection by *Penicillium* led to a rapid increase in the expression and activity of HmCDPK2, peaking at 12 h post-treatment [111]. In bananas, Panama disease, caused by *Fusarium oxysporum f.* sp. *cubense* Tropical Race 4 (Foc TR4), resulted in significant upregulation of *MaCDPK1-4* and *MaCDPK6* expression, with *MaCDPK2* and *MaCDPK4* showing the highest expression levels and increased sensitivity to Foc TR4 [30]. These findings suggest that MaCDPK2 and MaCDPK4 may play critical roles in banana resistance to Panama disease. Studies on F&Vs’ viral resistance have demonstrated that following tomato infection with tomato spotted wilt virus, 1022 proteins undergo significant changes, including the upregulation of CDPK and SnRK2, which are involved in activating the ABA signaling pathway [112].

These findings reveal the critical role of protein kinases in disease resistance and highlight their importance for improving disease resistance (Figure 1) and increasing yields of F&Vs.

## 4. The Mechanisms of Protein Kinases Regulate the Response of F&Vs to Stress

### 4.1. Regulation of Protein Kinases on the Antioxidant System

The growth and development of F&Vs are intricately regulated by a combination of internal and external factors. Under normal growth conditions, F&Vs generate ROS as part of their metabolic processes, playing essential roles in physiological functions such as cell signaling, hormone regulation, and cell wall synthesis [113]. However, exposure to adverse environments can result in excessive accumulation of ROS, leading to damage to the biological molecules and cell structures of F&Vs [114]. To counteract the oxidative damage caused by stress, F&Vs have evolved a sophisticated antioxidant system that includes a series of antioxidant enzymes, non-enzymatic antioxidants, and signaling molecules. Studies have shown that protein kinases participate in the stress response of F&Vs by regulating ROS metabolism. For example, in tomatoes, SlMAPK3 positively regulated the activities of superoxide dismutase (SOD), catalase (CAT), peroxidase (POD), ascorbate peroxidase (APX), and disease-related enzymes to inhibit ROS accumulation and enhance resistance to salt, drought, and gray mold [72,88]. Under cold stress, the overexpression of *SpCPK33* significantly increased the activities of SOD, POD, and CAT, suppressed ROS accumulation, and improved the cold tolerance of wild tomatoes [82]. Conversely, under heat stress conditions, mutations in *CPK28* resulted in ROS accumulation and increased protein oxidation while reducing the activities of APX and other antioxidant enzymes [95]. Additionally, SlMPK3 negatively regulated the heat tolerance of tomatoes by inhibiting the activity of the antioxidant defense system [44]. Under salt stress, VvMAPK9 in grapes enhanced the salt tolerance of callus tissues by increasing the activities of POD and SOD [32], whereas MdSOS2L1 in apples enhanced salt tolerance by boosting antioxidant metabolites, such as anthocyanins and malic acid [90]. In peaches, PpCDPK7 interacted with a respiratory burst oxidase homolog on the cell membrane, triggering the Ca^2+^-ROS signaling pathway to maintain cellular ROS homeostasis, thereby alleviating cold damage in fruit [37]. Additionally, ROS acts as signaling molecule that transmits stress signals to activate protein kinases. These kinases then regulate the transcription of various downstream genes through a series of phosphorylation events, modulating oxidative damage and coordinating the responses of F&Vs to various stressors [115]. These findings underscore the crucial role of protein kinases in regulating the antioxidant system and ROS metabolism to mediate the stress responses of F&Vs.

### 4.2. Interaction of Protein Kinases with Hormone Signaling Pathways

Phytohormones play crucial roles in the growth, development, and stress responses of F&Vs, with protein kinases exhibiting close and regulatory interactions with these hormones [116,117]. Protein kinases serve as intermediaries in hormone signaling, converting hormone signals into cellular physiological responses. Upon hormone binding to its receptor, specific protein kinases are activated, resulting in changes to their phosphorylation activities [118]. These activated kinases can then phosphorylate other proteins, initiating a cascade of signaling events. Additionally, protein kinases modulate various components of hormone signaling pathways by interacting with hormone receptors, regulating receptor activity and stability, and phosphorylating transcription factors within these pathways, thereby influencing gene expression.

For instance, overexpression of *SlMAPK3* in tomatoes could induce the expression of genes in the SOS pathway (*SlSOS1*, *SlSOS2*, *SlSOS3*) and the ethylene signaling pathway (*SlACS2*, *SlEIN2*, *SlERF2*), thereby contributing to the salt stress response of tomato plants [87]. ABA, a pivotal plant hormone that regulates growth, development, water balance, and stress responses, involves various protein kinases in signaling pathways [119]. For example, *NDW* in tomato plants positively regulated ABA synthesis and the expression of genes related to signal transduction, thereby promoting plant growth and cold resistance [96]. Heterologous expression of *PpSnRK1α* in tomato seedlings significantly increased the expression of genes related to ABA signaling, thereby improving salt tolerance [91]. In peppers, overexpression of *CaCIPK7* elevated the transcription levels of *CaNCED3* (a key gene in ABA synthesis) and downstream resistance genes (*CaRAB18*, *CaRD29B*, *CaDREB*), thereby enhancing drought tolerance [102]. In apples, overexpression of *MdMRLK2* upregulated ABA levels but downregulated SA levels. MdMRLK2 bound to the hypersensitive-induced response protein MdHIR1, inhibiting its action and negatively regulating apple resistance to *Valsa mali* [120]. Additionally, JA and methyl jasmonate (MeJA) are endogenous hormones and stress-signaling compounds widely distributed in plants [121]. Studies have demonstrated that JA can activate specific protein kinases to transmit signals that trigger diverse physiological responses within the cells. Knocking out *SlMAPK3* in tomato seedlings reduced the expression of the JA-synthesizing gene *SlLoxC* and downstream defense genes in its signaling pathway (*SlPI I* and *SlPI II*), whereas MeJA promoted the expression of *CaCIPK3*. Overexpression of *CaCIPK3* upregulated the expression of key genes in the JA synthesis and signaling pathways (*CaAOC* and *CaMYC*), thereby positively regulating drought resistance [101]. These studies together demonstrate that protein kinases modulate stress response processes in F&Vs by interacting with phytohormones.

### 4.3. Interaction of Protein Kinases with Transcription Factors

Transcriptional regulation is a fundamental process that coordinates normal plant development and the stress response. Transcription factors play vital roles in regulating gene transcription by specifically binding to cis-acting elements in gene regulatory regions [122]. Several studies have indicated that protein kinases are involved in the metabolic and stress response processes of F&Vs by regulating transcription factors. Protein kinases can interact directly or indirectly with transcription factors, leading to their phosphorylation. This phosphorylation can affect the transcription factors’ DNA-binding ability, subcellular localization, stability, and interactions with other proteins. Through this mechanism, protein kinases can modulate the function of transcription factors, thereby influencing gene expression. Conversely, transcription factors can regulate the expression levels of protein kinase genes, thereby affecting the production and activity of protein kinases. Under salt stress, increased osmotic stress leads to a rise in Ca^2+^ concentration, subsequently activate protein kinases, such as CDPKs and CIPKs. These activated protein kinases transmit osmotic stress signals to downstream transcription factors, including NAM, ATAF1/2, and CUC2 (NAC family), tryptophan-arginine-lysine-tyrosine (WRKY transcription factors), and v-myb avian myeloblastosis viral oncogene homolog (MYB family), which regulate gene expression to enhance the salt tolerance of F&Vs [123]. Cold stress triggered the phosphorylation of SlBBX17 by SlMPK1 and SlMPK2, promoting its interaction with the transcription factor HY5. This led to elevated HY5 levels and the induction of C-repeat binding factor gene expression, ultimately increasing the cold resistance of tomato seedlings [124]. Additionally, MusaMPK5 phosphorylated two stress-related NAC transcription factors, NAC042 and SNAC67, to improve the cold resistance of bananas [98]. In summary, the intricate interplay between protein kinases and transcription factors plays a crucial role in mediating the development and stress responses in F&Vs.

### 4.4. Regulation of Protein Kinases on Specific Proteins

The accumulation of heat shock response-related genes in F&Vs under high-temperature stress serves as a critical defense mechanism. Heat shock proteins (HSPs), including HSP100, HSP90, HSP70, HSP60, and small HSPs, are essential molecular chaperones that play crucial roles in maintaining protein homeostasis and ensuring plant survival under heat stress [125]. For instance, Yu et al. [28] demonstrated that SlMAPK3 negatively regulated heat tolerance in tomato plants. Notably, *SlMAPK3* mutants showed significantly higher transcript levels of *SlHSP70*, *SlHSP90*, and *SlHSP100* compared to wild-type plants, suggesting that SlHSPs may be involved in the SlMAPK3-mediated heat response. In addition to heat stress, disease resistance-related proteins such as SlMKK2 and SlMKK4 also play important roles in the immune response of tomatoes [109]. Protein-protein interaction assays and phosphorylation assessments indicated that tomato CPK27 directly interacted with and phosphorylated tonoplast sugar transporter 2, thereby regulating the accumulation of soluble sugars, which contributed to drought tolerance in tomato plants [99]. Furthermore, SlMAPK3 could also regulate drought tolerance in tomato plants by interacting with SlASR4 [104]. Under high-salt conditions, F&Vs experience disruptions in intracellular salt ion balance. In response, protein kinases regulate ion channels and proton pumps in both the plasma and vacuolar membranes. This regulation helps adjust ion concentrations in the cytoplasm, thereby regulating cellular osmotic pressure, controlling stomatal movement, and maintaining cellular homeostasis [126]. An imbalance in intracellular ion concentrations can cause necrosis of plant roots and leaves [127]. Overexpression of *SlCIPK24* could increase the deposition of Na^+^ and K^+^ in roots under salt stress by isolating excess Na^+^ in the vacuoles, thereby mitigating its detrimental effects in the cytoplasm [89].

These studies demonstrate the diverse mechanisms through which protein kinases regulate the stress response in F&Vs (Figure 2). However, the specific mechanisms by which they perceive and transmit certain stress signals require further investigation. Additional research is necessary to better understand how these mechanisms can be leveraged to improve the yield and quality of F&Vs.

## 5. Conclusions

Protein kinases are diverse and abundant, playing crucial roles in the growth, development, stress responses, and signal transduction processes of F&Vs. However, numerous protein kinases and their associated genes involved in the stress response of F&Vs have been identified; the focus has mainly been on MAPK and CDPK. Other protein kinases, such as PKA and PKC, which are more commonly studied in animals, remain less explored in plants. Moreover, although protein kinases function through multiple signaling pathways, there is a lack of comprehensive research on how these pathways interact when exposed to the same stimuli. Future investigations are warranted to gain a deeper understanding of the interactions between protein kinases and their specific regulatory mechanisms. In the future, advanced tools such as multi-omics integrated analysis, CRISPR-Cas9 gene editing technology, cell sequencing technology, fluorescence staining, and mass spectrometry can facilitate a comprehensive exploration of protein kinases in F&Vs. This exploration will reveal their regulatory mechanisms in plant metabolism, offering crucial theoretical insights into enhancing the quality characteristics of F&Vs.

## Figures and Tables

**Figure 1 ijms-26-00768-f001:**
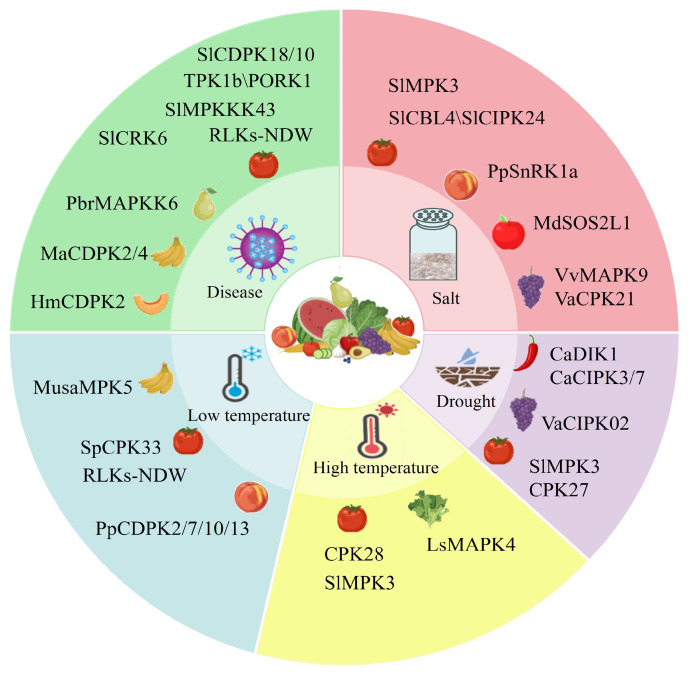
The role of protein kinases in various stress responses of F&Vs.

**Figure 2 ijms-26-00768-f002:**
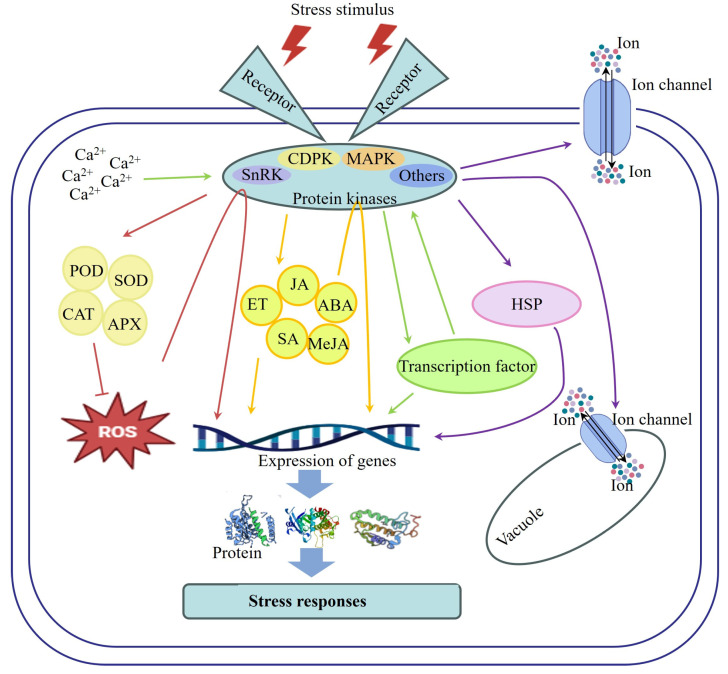
The mechanism of protein kinases in regulating the response of F&Vs to stress. Different-colored arrows represent diverse regulatory mechanisms: red represents the regulation of protein kinases in the antioxidant system; yellow represents the interaction of protein kinases with hormone signaling pathways; green represents the interaction of protein kinases with transcription factors; purple represents the regulation of protein kinases on specific proteins. Arrows indicate promotion, while lines with endlines indicate inhibition. Colorful points represent different ions passing through the ion channel.

**Table 1 ijms-26-00768-t001:** Protein kinases identified in F&Vs.

Species	Protein Kinase	Numbers	References
*Solanum lycopersicum*	AGC	17	[19]
	CDPK	29	[22]
	SnRK1	2	[27]
	SnRK2	8	
	SnRK3	30	
	MAPK	16	[28]
*Solanum habrochaites*	CDPK	33	[29]
*Solanum melongena*	CIPK	15	[30]
*Vitis vinifera*	CDPK	19	[31]
	MAPK	14	[32]
	SnRK2	8	[33]
	CIPK	20	[34]
*Capsicum annuum*	CDPK	31	[35]
	MAPK	19	[36]
*Prunus persica*	CDPK	17	[37]
*Musa nana*	CDPK	44	[38]
	SnRK2	11	[39]
*Musa acuminata*	MAPK	25	[40]
*Brassica rapa*	CDPK	41	[41]
	CIPK	51	[42]
	SnRK2	15	[43]
*Ananas comosus*	CDPK	17	[44]
	CIPK	21	[45]
*Fragaria ananassa*	SnRK1	1	[46]
	SnRK2	9	
	SnRK3	16	
	CDPK	11	[47]
	MAPK	43	[48]
*Fragaria vesca*	MAPK	12	[49]
	CDPK	19	[50]
*Citrullus lanatus*	MAPK	15	[51]
*Actinidia Chinensis*	MAPK	18	[52]
*Carica papaya*	MAPK	9	[53]
*Punica Granatum*	MAPK	18	[54]
*Morus alba*	MAPK	10	[55]
*Citrus reticulata*	CDPK	29	[56]
*Raphanus sativus*	CDPK	37	[24]
*Brassica napus*	CDPK	25	[57]
*Cucumis sativus*	CDPK	19	[58]
	SnRK1	1	[59]
	SnRK2	10	
	SnRK3	19	
*Cucumis melo*	MAPK	14	[60]
	CDPK	18	[61]
*Dioscorea opposite*	CDPK	29	[62]
*Brassica juncea*	CDPK	101	[63]
*Prunus avium*	SnRK2	6	[64]
*Malus prunifolia*	SnRK2	12	[65]
*Malus domestica*	MAPK	26	[66]
*Pyrus bretschneideri*	CIPK	28	[67]
*Dimocarpus longan*	CIPK	8	[68]

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
