# Peer review of "Advances in Protein Kinase Regulation of Stress Responses in Fruits and Vegetables"

_ijms, 2025, doi:10.3390/ijms26020768_

Round 1

Reviewer 1 Report

Comments and Suggestions for Authors

No comment

Comments on the Quality of English Language

no comment

Author Response

Thanks for improving our manuscript. In order to improve the quality of the manuscript, we have carefully revised the full text, including proofreading some expressions, tenses, and abbreviations.

Reviewer 2 Report

Comments and Suggestions for Authors

I find the paper well-written and interesting. However, I suggest reviewing the method to better explain the effect of protein kinase on stress responses because it is different for each vegetable and fruit, also suggest improving the references and using the latest published in the last three years.

Author Response

Comment: I find the paper well-written and interesting. However, I suggest reviewing the method to better explain the effect of protein kinase on stress responses because it is different for each vegetable and fruit, also suggest improving the references and using the latest published in the last three years.

Response: Thank you for your valuable advice! We have reviewed the method and added more detailed content to better explain the effects (line 192-195, 198-201, 237-239, 257-259). We have also cited the new references and incorporated additional detailed content into the article (line 223-226, 250-253, 281-285). In addition, based on the newly cited references, we have modified and supplemented the mechanisms of protein kinase involving in stress responses (line 406-412). The newly cited references are listed below:

  1. Wang, T, Z., Liu, M, J., Wu, Y., Tian, Y, F., Han, Y, Y., Liu, C, J., et al. Genome-wide identification and expression analysis of MAPK gene family in lettuce (Lactuca sativa) and functional analysis of LsMAPK4 in high temperature-induced bolting. Int. J. Mol. Sci. 2022, 23, 11129. doi: 10.3390/ijms231911129.
  2. Zhu, C, A., Jing, B, Y., Lin, T., Li, X, Y., Zhang, M., Zhou, Y, H., et al. Phosphorylation of sugar transporter TST2 by protein kinase CPK27 enhances drought tolerance in tomato. Plant Physiol. 2024, 195, 1005-1024. doi: 10.1093/plphys/kiae124.
  3. Dong, X, N., Lu, H, M., Zhao, L, Q., He, B, Q., Zhang, J, J., Zhao, B., et al. SlMAPKKK43 regulates tomato resistance to gray mold. Acta Hortic. Sin. 2024, 51, 309-320. doi: 10.16420/j.issn.0513-353x.2023-0757.

Furthermore, we have redrawn Figure 1 to enhance its clarity and visual appeal, ensuring it better illustrates the roles of different protein kinases in various fruits and vegetables under stress conditions (line 301). In order to improve the quality of the manuscript, we have carefully revised the full text, including proofreading some expressions, tenses, and abbreviations. Additionally, we have updated the references by citing some new references from the past three years.

Reviewer 3 Report

Comments and Suggestions for Authors

This manuscript introduced and summarized types of protein kinase in fruits and vegetables, listed their potential role and regulation in response to abiotic (including salt, drought, and temperature stress) and biotic stresses. Although it would be good to provide more details of regulatory up-/down-stream targets instead of linking mutation/overexpression of protein kinase genes directly to phenotype when giving examples, I believe it can be considered for acceptance after a minor issue is fixed: Figures should be referred in the main text.

Author Response

Comment: This manuscript introduced and summarized types of protein kinase in fruits and vegetables, listed their potential role and regulation in response to abiotic (including salt, drought, and temperature stress) and biotic stresses. Although it would be good to provide more details of regulatory up-/down-stream targets instead of linking mutation/overexpression of protein kinase genes directly to phenotype when giving examples, I believe it can be considered for acceptance after a minor issue is fixed: Figures should be referred in the main text.

Response: Thanks for your suggestions. Figures 1 has been referred in the correct text (line 214, 247, 262, 299), and Figures 2 has also been appropriately cited in the correct text (line 422). Furthermore, we have redrawn Figure 1 to enhance its clarity and visual appeal, ensuring it effectively illustrates the roles of different protein kinases in various fruits and vegetables under different stress conditions (line 301).